# Role of Epitranscriptomic and Epigenetic Modifications during the Lytic and Latent Phases of Herpesvirus Infections

**DOI:** 10.3390/microorganisms10091754

**Published:** 2022-08-30

**Authors:** Abel A. Soto, Gerardo Ortiz, Sofía Contreras, Ricardo Soto-Rifo, Pablo A. González

**Affiliations:** 1Millennium Institute on Immunology and Immunotherapy, Departamento de Genética Molecular y Microbiología, Facultad de Ciencias Biológicas, Pontificia Universidad Católica de Chile, Santiago 8330025, Chile; 2Millennium Institute on Immunology and Immunotherapy, Laboratorio de Virología Molecular y Celular Programa de Virología, Instituto de Ciencias Biomédicas, Facultad de Medicina, Universidad de Chile, Santiago 8380492, Chile

**Keywords:** herpesvirus, latency, RNA modifications, DNA modifications, m6A, 5mC, m5C, pseudouridinilation, 2′-O-me, histone modifications

## Abstract

Herpesviruses are double-stranded DNA viruses occurring at a high prevalence in the human population and are responsible for a wide array of clinical manifestations and diseases, from mild to severe. These viruses are classified in three subfamilies (*Alpha-*, *Beta-* and *Gammaherpesvirinae*), with eight members currently known to infect humans. Importantly, all herpesviruses can establish lifelong latent infections with symptomatic or asymptomatic lytic reactivations. Accumulating evidence suggest that chemical modifications of viral RNA and DNA during the lytic and latent phases of the infections caused by these viruses, are likely to play relevant roles in key aspects of the life cycle of these viruses by modulating and regulating their replication, establishment of latency and evasion of the host antiviral response. Here, we review and discuss current evidence regarding epitranscriptomic and epigenetic modifications of herpesviruses and how these can influence their life cycles. While epitranscriptomic modifications such as m^6^A are the most studied to date and relate to positive effects over the replication of herpesviruses, epigenetic modifications of the viral genome are generally associated with defense mechanisms of the host cells to suppress viral gene transcription. However, herpesviruses can modulate these modifications to their own benefit to persist in the host, undergo latency and sporadically reactivate.

## 1. Introduction

Herpesviruses are double-stranded DNA viruses with genomes that range between 152–172 kbp in length. These viruses are members of the *Herpesviridae* family, which encompasses at least 100 different viruses [1]. From these, eight herpesviruses have been shown to affect humans, which are distributed within three subfamilies: *Alpha-, Beta-* and *Gammaherpesvirinae* [2]. While herpes simplex viruses type 1 and 2 (HSV-1 and HSV-2, respectively) and the varicella-zoster virus (VZV) are classified within the *Alphaherpesvirinae* subfamily, cytomegalovirus (CMV) and the human herpesviruses 6 and 7 (HHV-6, HHV-7) belong to the *Betaherpesvirinae* subfamily, and Epstein Barr virus (EBV) and the Kaposi’s sarcoma-associated herpesvirus (KSHV) are members of the *Gammaherpesvirinae* subfamily [2]. All these viruses can infect humans asymptomatically or symptomatically with a variety of symptoms and clinical manifestations, leading in some circumstances to mild up to severe diseases. Other commonly studied herpesviruses that affect animals are pseudorabies virus (PRV) and Marek’s disease virus (MDV), both belonging to the *Alphaherpesvirinae* subfamily [3,4].

## 2. Herpesviruses: Epidemiology and Illnesses

Human viruses of the *Alphaherpesvirinae* subfamily, such as HSV-1 and HSV-2, are highly prevalent worldwide with a prevalence estimated at 66.6% and 13.2%, respectively [5]. Data gathered in Asia indicate that these viruses have a frequency of 50% in children and 76.5% in adults [6], while in other countries lower values have been reported, such as 47.1% for HSV-1 and 12.1% for HSV-2 in the USA [7]. Diseases generated by HSV-1 and HSV-2 are usually mucocutaneous or affect the central nervous system [8]. The main transmission route of HSVs is through oral, oro-sexual, or sexual contact [8]. During early stages of infection, viral glycoproteins interact with their ligands on the cell surface for viral binding. HSV virions are overall composed by a lipid bilayer envelope decorated with numerous proteins and glycoproteins on the outer surface. Beneath the viral envelope, there are approximately 20 viral proteins, in a zone called the tegument, which harbors several viral proteins that play essential roles over the host antiviral response early on after infection [9]. Below the tegument is the viral capsid which harbors the viral genome in a lineal form [9]. A similar structure is found in other herpesviruses, although with particularities for each of them. At least five glycoproteins are critical for HSV-1 entry and four for HSV-2 [10]. Glycoprotein B (gB) is key for virus attachment to the heparan sulfates on the host cell for both HSVs, while gC is also important for HSV-1, but not for HSV-2 [11]. gD is also present on the virion surface, and binds to any of the receptors mentioned above, namely nectin-1, nectin-2, the herpes virus entry mediator (HVEM), or 3-OS heparan sulfates [10,12]. Upon binding to one of its ligands gD activates two other essential glycoproteins on the virion surface, gH and gL, which act as a complex (gH/gL) that in turn activates the fusogenic properties of gB [10]. Once these interactions are established, HSV entry into the cytoplasm is mediated by membrane fusion, endocytosis or phagocytosis [13,14]. The viral capsid then enters the cell and is transported through microtubules to the nuclear pore, where the genome is injected into the nucleus and viral gene transcription initiates [15]. At this stage, productive viral infection begins with the transcription of a particular set of genes named immediate early genes (IE or alpha) [16]. A pivotal transcription factor called ICP4 (infected cell protein 4) is transcribed and translated early on (referred to as an immediate early gene; IE or alpha gene), which allows the beginning of a downstream sequential gene expression cascade that promotes the transcription and translation of early genes (E or beta genes), such as ICP8 (infected cell protein 8) or TK (thymidine kinase), and finally late genes (L or gamma genes), such as VP16 (viral protein 16) or gD (glycoprotein D) [16]. The latter are sometimes further classified as early late (gamma-1) and late (gamma-2) genes [17]. The classification of IE, E and L genes is based on the timing of their transcription during infection and dependency on different viral transcription factors [14,18].

Varicella-zoster virus (VZV), another alphaherpesvirus, causes varicella (also known as chickenpox) and zoster (also known as shingles) [19]. Current evidence suggests that primary VZV infection begins with the replication of the virus in respiratory epithelial cells [20]. Similar to other alphaherpesviruses, VZV binds to heparan sulfate receptors on the cell surface for its entry into the cell. Moreover, VZV can use the insulin-degrading enzyme (IDE) as the cellular receptor through an interaction that is mediated by the viral glycoprotein E (gE) in the MeWo cell line, which is a model of fibroblast. Once the virus is inside the cell, the expression of IE, E and L genes is initiated [21]. Infection is followed by a widely distributed vesicular rash that is a typical trait of varicella after the incubation period (approximately 10 to 21 days) [19]. This pattern likely reflects viral spread from the tonsils and other local lymphoid tissues, from where infected T cells can transport the virus via the bloodstream to the skin [22]. VZV is found in a worldwide geographic distribution, but annual epidemics are more prevalent in temperate climates, occurring most often during late winter and spring [23]. The prevalence of VZV infection in the population is elevated and, in the USA, has been reported to be over 93% in the 6- to 19-year old group and 98% in adults aged 20–49 years [24,25]. Cases of herpes zoster provide a source of VZV transmission to susceptible close contacts, causing varicella; the virus then spreads rapidly to other susceptible individuals, in part because, in contrast to other herpesviruses, VZV is transmissible by the respiratory route [19]. Varicella attack rates among susceptible household contacts exposed to VZV are approximately 90%; more limited exposures, such as those occurring in school classrooms, result in transmission rates of about 10 to 35% [26].

Two other alphaherpesviruses of interest are pseudorabies virus (PrV) and Marek’s disease virus (MDV), which can infect animals. PrV is a neurotropic virus that causes Aujeskzy’s disease in swine, although a broad spectrum of mammals can be infected with this virus, such as rodents, cats, dogs and cattle [27]. The seroprevalence of PrV in pigs in China has been reported to be approximately 35% [28]. Similar to HSV-1, neuron infection with PrV occurs through trigeminal neuron infection involving the interactions of the viral glycoprotein gD with the host nectin-1 receptor, which activates cytoskeleton remodeling, allowing viral particle entry into the cell [29]. MDV infection on other hand causes one of the most prevalent cancers described in the animal kingdom and mainly poultry [30]. The prevalence of infection can reach up to 49.5% in chicken, as reported in Iraq [31]. MDV infection is mediated by the interaction of the viral glycoprotein gC with glycosaminoglycans (GAGs) on cell surfaces and this virus mainly infects and replicates in immune cells, such as macrophages, natural killer (NK cells), as well as B cells, and establishes latency in T cells [30].

Betaherpesviruses such as CMV have a high prevalence in the infant population, with approximately 59% of children older than six years old having been exposed to the virus [32]. Strong differences in the epidemiology of CMV amongst the adult population in industrialized and emerging countries exist, where the latter have the highest frequency of infection reaching 100%, while industrialized countries display a range of infection between 60–70% [33]. CMV transmission occurs by direct contact with infected body fluids and there is evidence regarding vertical and breast-milk transmission [33]. The main cells infected by CMV are epithelial cells, fibroblasts and myeloid cells [34]. The mechanism of entry described for CMV involves molecular interactions between viral glycoprotein complexes, such as gH/gL and gH/gL/gp42, with integrins on the surface of epithelial cells and major histocompatibility complex class II (MHC-II) molecules in B cells [35]. Symptoms associated with CMV infections vary significantly, from asymptomatic to severe illness, with mononucleosis being the most common clinical presentation [36].

Another betaherpesvirus that can infect humans is HHV-6 (also known as HHV-6A). This virus infects children at a high rate and 90% of the human population is likely to being infected before the age of three [37]. The general route of infection is by contact with contaminated saliva, after which the virus enters mainly into CD4^+^ T cells through the CD46 receptor [38]. Symptoms after HHV-6 infection depend on the immunological state of the host. Infection in immunosuppressed individuals is related with the development of ataxia, hypersomnia, dementia, encephalitis, and myocarditis, among others [37]. The human herpesvirus 7 (HHV-7, also known as HHV-6B), is another betaherpesvirus able to infect humans and is highly prevalent worldwide [39]. In the United States of America, the prevalence of HHV-7 is above 85% [40]. However, prevalence rates can significantly vary between ethnicities [40]. This virus enters to the target cells through the CD134 receptor which is expressed only in activated CD4^+^ T cells [41]. Usually, HHV-7-related symptoms after infection only occur in immunosuppressed individuals and, in these subjects, they are somewhat similar to those reported for HHV-6 [39].

Epstein Barr virus and the Kaposi’s sarcoma-associated herpesvirus (KSHV, or HHV-8) are gammaherpesviruses that can infect humans [42]. EBV infection in the population is widespread, over 90%, with asymptomatic infections mainly occurring during childhood [43]. EBV mainly infects B cells through the interaction between the viral glycoprotein-350 (gp-350) and the B cell receptor CD21 [44]. If infection occurs during adulthood, it can result in infectious mononucleosis, a self-limiting disease characterized by the inflammation of lymph nodes in the neck region [43]. Transmission of EBV is mainly through oral secretions [42], although infections after blood transfusions and organ transplants also occur [42]. EBV is known to be an oncogenic virus, as its infection is associated mainly with Hodgkin’s lymphoma [45]. Additionally, EBV reactivation in immunosuppressed individuals may cause lymphoproliferative diseases [45]. On the other hand, recent evidence links EBV with the development of multiple sclerosis [46].

As its name suggests, KSHV is responsible for the development of all Kaposi’s sarcomas [47]. Similar to other types of herpesviruses, KSHV interacts with heparan sulfates on the cell surface to enter the cell through the use of its viral glycoprotein gB [48]. In children, cases of hemophagocytic lymphohistiocytosis are described after infection with this virus [49]. The epidemiology of KSHV significantly depends on the geographical region [50]. While the prevalence of this virus in North America, Europe and Asia is lower than 5%, in Eastern Europe, the Middle East and the Caribbean the seroprevalence is above 50% [50]. Noteworthily, the highest seroprevalence of KSHV is found in Africa and regions of the Amazonas in Brazil, with values reaching 60% [50]. While the transmission route for this virus is not totally clear, its high prevalence in men that have sex with men supports the hypothesis of sexual transmission [50]. However, there is a high level of seroprevalence in children in Africa ,also suggesting salivary transmission [50]. Vertical and blood transfusion transmission are also well described [51]. Importantly, this virus is able to infect numerous cell types, such as dendritic cells (DCs), monocytes, B lymphocytes and oral epithelial cells [48,51].

Overall, herpesviruses are present in humans at a high prevalence, with the exception of HHV-8, which has a more limited penetration in the population. Importantly, these viruses elicit lifelong infections through the establishment of persistent infections and can undergo latency. Lifelong infections and frequent reactivations for some of these viruses evidences the existence of immune evasion mechanisms that allow them to persist in the host. Although there are numerous antivirals available that limit severe diseases caused by these viruses, they do not clear the virus from the host and thus, reactivations may occur during the entire life of an infected individual. Furthermore, some clinical manifestations elicited by these viruses, such as skin lesions produced by HSV-1 and HSV-2, are only mildly affected by such drugs, reducing in only a few days what are usually week-long lesions [52]. Therefore, a better understanding of the molecular determinants that favor herpesvirus infections and their persistence in the host is needed for developing better therapeutic approaches. In this review we revise and discuss common epitranscriptomic and epigenetic modifications reported in virus-derived nucleic acids and how these are modulated by host and viral determinants.

## 3. Epitranscriptomic Modifications and Herpesviruses


**N^6^-methyl-6-adenosine (m^6^A):**


The N^6^-methyladenosine (m^6^A) modification of RNA molecules consists of the addition of a methyl group at position N^6^ of the adenosine nucleotide and is the most prevalent internal modification occurring in mRNA [53]. The reaction is catalyzed by enzymes called “writers” and takes place co-transcriptionally in the nucleus [54]. Deposition of m^6^A onto mRNA molecules is catalyzed by a methyltransferase complex formed by METTL3 and METTL14, which uses S-adenosylmethionine (SAM) as a substrate [55]. Additionally, METTL3/METTL14 complex is associated with the protein WTAP, which is important for substrate targeting [55]. On the other hand, a set of “reader” proteins recognize m^6^A-modified mRNA molecules to exert the function of the modification [56]. The best characterized m^6^A reader proteins belong to the YTH-domain family (YTH) and are associated with the regulation of different steps of mRNA metabolism ranging from alternative splicing, nuclear export, translation efficiency, to degradation and subcellular localization [57,58,59,60]. Lastly, other enzymes that interact with RNAs having m^6^A modifications are the “erasers” such as alkb homolog 5 RNA demethylase (ALKBH5) and alpha-ketoglutarate-dependent dioxygenase (FTO), which can reverse this modification through demethylation reactions [54].

Importantly, the m^6^A modification in RNA has been widely described in the context of viral infections amid both RNA and DNA viruses, including SARS-CoV-2, HIV-1, Zika, dengue, hepatitis C virus (HCV) and HSV-1, among others [61,62,63,64,65,66]. The role of the m^6^A RNA modification has been related to different processes occurring during the viral replication cycle, including viral gene expression and assembly of new virions, but also immune evasion [67].


**Pseudouridine:**


Pseudouridine (Ψ) is the most abundant internal RNA modification found in transfer RNA (tRNA) and ribosomal RNA (rRNA) [68]. This modification is irreversible and occurs through two different pathways, one that is RNA-dependent and another that is RNA-independent. RNA-dependent mechanisms involve a family of ribonucleoproteins called H/ACA small nuclear ribonucleoproteins (sRNPs) that use a guide RNA (gRNA) complementary to the sequences that will be modified. This gRNA hybridizes with target sequences and adopts a secondary structure so that the sRNPs recognize this structure and can catalyze the isomerization of uracil into pseudouridine [68]. RNA-independent mechanisms involve pseudouridine synthase enzymes (PUS) [68], which can recognize secondary structures within RNA molecules that contain uridine targets [68]. These enzymes are known to modify tRNA (PUS 2/3/4/6/9), small nucleolar RNA (snRNA) (PUS 1/7), rRNA (PUS 5/7) and mRNA (PUS 1/2/3/4/6/7/9). Interestingly, reader proteins related to this modification have not been identified to date [69].

Importantly, pseudouridylation is known to occur in both tRNA and rRNA, yet its role in mRNA is unclear to date [70]. Current evidence suggests that pre-mRNA pseudouridylation can modulate alternative splicing through diverse mechanisms, such as the modification of the affinity of RNA-binding proteins to RNA, or by modulating secondary structures within pre-mRNAs [70]. Pseudouridylation might stabilize RNA, as it can modify its structure by eliciting base-pairings that promote a rigid backbone [71]. Additionally, there is evidence suggesting that pseudouridylation confers RNA resistance to degradation, as there are studies reporting that pseudouridine-modified RNAs are more resistant to snake venom and spleen phosphodiesterases, as compared to RNAs with non-modified uridine [72]. Another process in which pseudouridylation is involved is during translation, as pseudouridylation in mRNAs can be recognized by the MetRS (which is an tRNA synthetase of methionine specifically), for its translation with regulatory purposes [73]. This may reduce the binding of antisense RNA molecules improving translation efficiency [73]. An interesting finding regarding pseudouridylation is its role in immune escape. Recent evidence suggests that a complete replacement of uridine with pseudouridylation dramatically decreases the levels of IFN-β mRNA due to RNA containing pseudouridylation, failing to activate conformational changes in retinoic acid-inducible gene I protein (RIG-I), which is directly related to the activation of the antiviral interferon pathway [74,75,76,77]. While RNA molecules carrying uridine modifications can activate some Toll-like receptors, such as TLR3, to switch on host immune responses [76], pseudouridylation is not recognized by such TLRs due to steric incompatibility [76].

Importantly, pseudouridylation modifications have been described in viral RNAs, such as HIV-1 and SARS-CoV-2 [78,79]. For example, in SARS-CoV-2 pseudouridylation was detected in mRNA, yet its role is unclear and the question of whether pseudouridylation may be associated with immune escape remains open [79]. Taken together, more studies are needed to better understand the effect and role of pseudouridylation in the context of viral infections. Interestingly, pseudouridylation modifications have been used in the development of mRNA-based vaccines due to its characteristics that are related to a decreased activation of anti-RNA innate immune system components [72] and increased production of the antigen [72].


**Methyl-5-cytosine (m^5^C)*:***


The 5-methylcytosine modification in RNA molecules consists in the addition of a methyl group to the carbon in position 5 of cytosine, which is catalyzed by methyltransferase enzymes. To date, NOL1/NOP2/SUN domain (NSUN) proteins (NSUN 1–7) have been shown to introduce such a methyl group in mRNAs [80]. Interestingly, m^5^C has been identified in mRNA molecules isolated from diverse types of viruses, such as SARS-CoV-2 and HIV-1, as well as in non-coding RNA of the herpesvirus EBV [81,82,83]. These modifications have been reported to be related with both the function and stability of mRNA molecules and have been found in at least 1.4% of transcripts derived from retroviruses. Additionally, occurrence is ten-times higher in these mRNAs than in cellular mRNAs during infection [82]. Noteworthily, m^5^C has been associated with an increase in the translation of HIV-1 mRNAs [84].


**2′-O-Methylation (2′-O-me):**


The 2′-O-methylation modification in RNA molecules consists in the addition of a methyl group in the 2′-OH of the ribose [85]. This modification has been identified in rRNA and tRNA molecules and evidence suggests that it can also be found in mRNA transcripts [85]. Importantly, all nucleotides, either canonical or non-canonical, can be 2′-O-methylated by sets of enzymes called MTases [85].

In the cellular context, 2′-O-Methylation is mainly associated with RNA stability and translational modulation [85]. A role for this modification in viral infections has been described for the West Nile virus, in which the 2′-O- methylation of the 5′ cap enhances its virulence in macrophages, because this modification allows the viral evasion of intrinsic cellular defense mechanisms, such as interferon-induced proteins with tetratricopeptide repeats (IFIT), which are associated with antiviral activity. These proteins inhibit translation by interactions between their tertiary structures and partner proteins used by viruses to translate their proteins, such elongation factors eIF2, eIF3 and eIF4 [86,87]. Cap 2′-O- methylation has been described during SARS-CoV-2 infection, with this modification being performed by the nonstructural proteins NSP10/14 and NSP10/16 heterodimers. Interestingly, this allows the virus to evade host nucleases and protects mRNA from degradation, providing the virus additional immune escape capabilities [88,89]. Moreover, internal 2′-O-methylation was shown to avoid immune sensing of the HIV-1 RNA in dendritic cells and macrophages [90].

Overall, RNA modifications in the context of viral infections are diverse, with multiple different nucleotides being affected and different effects over the stability and functions of RNA molecules (Table 1). For instance, some of these modifications can enhance viral mRNA stability and prevent degradation or enhance the efficiency of mRNA translation. Moreover, these modifications can be helpful for the virus for evading the host antiviral immune response by affecting the recognition of viral RNAs by Toll-like receptors and RIG-I-like receptors (RLRs), therefore dampening the activation of the cellular antiviral signaling pathways. Whether these modifications occur or not in the context of viral infections with herpesviruses and their roles are discussed below.

### 3.1. Episodic Splicing Modifications

Constitutive splicing is an RNA maturation step that consists of the removal of introns and the ligation of exons within pre-mRNA molecules. This process is mediated by small nuclear ribonucleoproteins (i.e., U1, U2, U3, U4, U5 and U6), which form a complex known as the spliceosome [91]. Importantly, the cleavage of introns and the ligation of exons can vary depending on the RNA molecule and eventually generate different types of mRNA molecules starting from the same transcript in a process called alternative splicing [91].


**Alternative Splicing Events**


Alternative splicing can be generated by the inclusion or exclusion of introns and exons in the final mRNA transcript [92]. Among alternative splicing events, exon skipping is the most common, which consists of the inclusion or exclusion of an exon into the mature mRNA molecule. While the exact mechanism of exon skipping is somewhat unclear, there is evidence that relates this process with nonsense mutations, although additional investigations are needed [92]. Another alternative splicing process consists of mutually exclusive exons in which the splicing of exons is performed in a coordinated manner, where one exon is retained and another is spliced-out. Finally, intron retention occurs when a complete and unspliced intron is retained within the mature mRNA [91]. Other methods generating alternative mRNA transcripts consist of the use of alternative transcription start sites (TSS), as displacing the transcription initiate site can lead to the exclusion of exons [92].


**Splicing Modifications and Herpesvirus Infection**


For HSV-1, very few genes have been described to undergo splicing. However, there is evidence for a role of the viral protein ICP27 in regulating alternative splicing during infection. Indeed, ICP27 has been reported to inhibit alternative splicing during lytic infection of human epithelial kidney cells (HEK-293 cells) [93]. Interestingly, the inhibition of alternative splicing has been described to be mediated through the interaction of ICP27 with snRNPs involved in this process, which were redistributed [91]. Conversely, when ICP27 is absent during the latent phase of HSV-1, viral transcripts undergo splicing to restrict viral antigen expression and evade the host immune response [93]. Alternative splicing has also been described for MDV [94]. In this case, the ICP27 protein of MDV can interact with splicing regulator proteins (SR) in lymphoid cells (MSB-1 cells, which is the lymphoblastoid cell line) [91]. However, as opposed to what has been described with HSV-1, the role of MDV ICP27 in alternative splicing of viral transcripts has not been reported, and thus it remains to be clarified if this protein has any impact over the virus’ lytic or latent phases. Regarding PRV, alternative splicing events have been described in mRNA transcripts of the *epo* and *ul21* genes in epithelial cells (i.e., PK15 cell line) [95,96]. However, similar to MDV, the role of alternative splicing events during the latent or lytic phases of this virus remains an important question to address experimentally.

For the betaherpesvirus CMV, an interesting role for alternative splicing has been described, because alternative splicing in the major immediate early gene locus (MIE), particularly in exon 4, generates an abortive infection in human foreskin fibroblast cells. Therefore, alternative splicing of MIE during the lytic phase could be detrimental for viral replication [97].

Regarding the gammaherpesvirus EBV, there is evidence indicating that EBNA-2 and EBNA-LP viral proteins can modulate alternative splicing and regulate the expression of genes related with the survival and proliferation of infected B cells, as B cell lymphoma protein (Bcl-L) and Numb endocytic adaptor protein (NUMB); both are regulated by alternative splicing [98]. For KSHV, the alternative splicing of viral transcripts has been described for ORF57, which promotes the expression of lytic genes and can regulate the splicing of the biscistronic ORF70/K3 RNA, which is associated with viral DNA replication in the lytic phase of this virus and immune evasion [99].

Taken together, alternative splicing of viral RNAs in herpesvirus has been reported and relates to immune escape and optimal viral lytic phases, although further studies are needed to fully grasp all the viral and host factors involved and how they are modulated during both the lytic and latent phases of viral infection.

### 3.2. Episodic Transcriptional Modifications in Herpesvirus Infections


**Epitranscriptomic Modifications in Alphaherpesvirus Infections**


Overall, the mechanisms by which m^6^A contributes to viral infection with HSV-1 are poorly understood and evidence is contrasting [65]. At present, there are studies that report that m^6^A modifications do exist in HSV-1 transcripts in host cells, such as human epithelial cells (HeLa), and this modification is critical for viral replication [65]. Interestingly, infection by HSV-1 increases the expression of m^6^A writers METTL3 and METTL14 in infected cells [65]. Moreover, silencing of these proteins results in a significant reduction in viral replication [65]. Consistently, an enhancement in viral replication and replication was observed when the eraser enzyme ALKBH5 was knocked-down [65]. However, recent evidence suggests that the alphaherpesvirus protein US3 of the pseudorabies virus (PRV), which has kinase activity, can inactivate the m^6^A writer complex METTL3/METTL4/WTAP through phosphorylation in a fibroblast cell line (ST), producing a decrease in m^6^A levels and evidencing that m^6^A seems not to be important for PRV replication [100].Therefore, effects of m^6^A in viral replication may be specific for the cellular type infected and the alphaherpesvirus evaluated. Noteworthy, it has been observed that upon HSV-1 infection of fibroblasts, the m^6^A writer METTL3 is re-distributed from the nucleus to the cytoplasm. Importantly, these changes in the distribution of this protein were mediated by the HSV-1 protein ICP27, particularly by the carboxy-terminus domain of this protein which has both structural and sequence homology with regulators of RNA processing and export [101].

Regarding m^6^A reader proteins, these have also been found to be important in the replication of HSV-1 by unknown mechanisms. Given the generally described effects of m^6^A over the increase in stability of mRNAs and translation efficiency, HSV-1 may recruit writer proteins to its modified mRNAs to improve their stability and translation efficiency, which would likely produce enhanced viral replication [65,101].

For other alphaherpesviruses, such as MDV, infection has been reported to modulate the m^6^A landscape in host cells, mainly by increasing the frequency of the m^6^A modification in long non-coding RNAs (lncRNAs) in a chicken fibroblast cell line [102]. However, additional evidence is needed to understand how these changes could affect the replication of this virus.

Noteworthily, the effects of m^5^C, Ψ and 2′-O-methylations in the context of alphaherpesvirus infections have not been reported to date (Figure 1). Nevertheless, during recent years new technologies, such as direct RNA sequencing, have been emerging, which may help to elucidate any effects these modifications may have in the replicative cycle of alphaherpesviruses relevant to human health, such as HSV-1, HSV-2 and VZV [103].

Given these methodological advances, we foresee in the near future the identification and characterization of pseudouridylations and 2-O-methylations within alphaherpesvirus infections.


**Epitranscriptomic Modifications in Betaherpesvirus Infections**


CMV infections have been associated with an increase in reader proteins, such as YTHDF2 in NK cells, and inhibiting this protein in vivo reduced the host’s capacity to control this virus [104]. Interestingly, CMV infection of NK cells affects proteins that read or erase the m^6^A modification in the host cell. For example, an increase of the YTHDF2 reader protein was observed 1.5 days post infection, which was accompanied by a decrease in YTHDF3. Increased YTHDF2 expression was observed in response to viral infection and was associated with the activation of the transcription-factor signal transducer activation of transcription 5 (STAT-5), which activates host antiviral pathways mediated by IFN-γ [104]. However, this response may vary depending on the cell type, as an increase in the expression of the writer proteins METTL3, METTL14, together with the YTHDF2 reader, was observed after CMV infection in fibroblasts. Noteworthily, these enzymes were found to be necessary for virus propagation [105]. Indeed, when the writer protein METTL3 was inhibited in fibroblasts infected with CMV, an enhanced type-I interferon response was observed, suggesting that CMV may use this modification as an evasion mechanism [105].

Findings from another study propose that CMV uses the host m^6^A RNA modification machinery to modify the expression of host proteins during infection. Zu and colleagues. observed that CMV infection in endothelial cells elicits an aberrant and increased expression of writer enzymes, such as METTL3, and YTHDF2 reader proteins. This effect produced a modification of mRNA encoding host proteins, such as MCU (mitochondrial calcium uniporter), with METTL3 modifying the mRNA of this host factor in the 3′ UTR region, which in turn was read by YTHDF2, resulting in an aberrant expression of MCU that facilitates apoptotic processes within the cell [106]. Similar to alphaherpesviruses, further evidence regarding the role of other RNA modifications in betaherpesvirus infections require studies yet to be carried out and reported (Figure 1).


**Epitranscriptomic Modifications in Gammaherpesvirus Infections**


The effects of the m^6^A modification have also been studied in EBV. Interestingly, the EBV early protein BZLF1 was shown to bind the promoter of the writer enzyme METTL3, negatively modulating its expression [107]. Moreover, Zheng et al. further described that during the pre-latency phase of EBV infection, the expression of the viral protein EBNA-2, which is involved in transcriptional regulation of viral transcripts and is associated with the immortalization of EBV-infected cells [108], decreased upon knock-down of METTL3 in B cell lymphoma cells (BJAB cells) [109]. An important observation reported in this work was that EBV infection can induce the hypomethylation of the mRNA of TLR9, affecting its stability, which may affect the initiation of an innate immune response, altogether favoring the virus’ capacity to achieve long-term latency in the host [109]. Another study found that the deletion of the writer protein METTL3 in B cells negatively affected the expression of viral lytic proteins and decreased the yield of progeny virions, further suggesting that METTL3 may have a role in the replicative cycle of EBV [110].

Another modification that has been studied in the context of EBV infection is m^5^C. Some studies suggest that the EBV viral non-coding RNA of the EBER1 gene is modified with m^5^C [81], and that the writer enzyme that performs this modification is NSUN2, which is associated with a decrease in EBER1 levels [81,110]. However, the non-coding RNA EBER can bind to Toll-like receptors and activate the interferon pathway, hence the inhibition of EBER1 levels in the host cell, due to m^5^C modifications maybe acting as an immune escape mechanism evolved by this virus [81].

Another gammaherpesvirus undergoing the m^6^A modification in its transcripts is the Kaposi’s sarcoma-associated herpesvirus (KHSV), whose viral mRNA is extensively methylated with m^6^A during the lytic and latent phases, with a marked increase in mRNA displaying the m^6^A modification when lytic replication is stimulated [111]. Importantly, enzymes involved in this modification are critical for KSHV infection, with evidence suggesting that the reader protein YTHDF2 suppresses KSHV infection. This suppressive effect has been reported to be mediated by the deadenylation and degradation of viral transcripts [88]. However, findings in this area are contrasting and effects of writer and reader proteins apparently depend on the cell type used [112]. For example, Hesser and colleagues. showed that YTHDF2 and METTL3 are necessary for virion production in Kaposi’s sarcoma-derived cells (iSLK.BAC16 cells). Yet, the effects were different in other cells obtained from KS, such as iSLK.219 cells, in which YTHDF2 and METTL3 were associated with the accumulation of the viral lytic transactivator ORF50, and not in virion production [112]. Moreover, the non-YTH reader protein staphylococcal nuclease and tudor domain containing 1 (SND1) has been reported to be involved in the replication of KSHV and its disruption found to interfere with the replication cycle of KSHV in a cell line of B-cells (TREx BCBL1-Rta cells) due to the protective role of SND1 in ORF50 mRNA, stabilizing and protecting against degradation [113]. ORF50 is the first lytic protein produced by this virus and is necessary to switch between lytic and latent phases [113].

Overall, the effects of m^6^A during KSHV indicate that this modification is necessary for viral replication, and that virus infection per se can modify the landscape of viral and host m^6^A RNA modifications. However, similar to the infection with CMV, the reader protein YTHDF2 has also been involved with an antiviral activity; in this case specifically mediating viral transcript degradation. On the other hand, YTHDF3 would seem to have a positive role in viral replication through a mechanism yet to be determined. The influence of other RNA modifications during both the latent or lytic phases of this virus remains to be identified (Figure 1).

## 4. Epigenetic Modifications during Viral Infections


**Histone Modifications**


Some viruses containing double-stranded DNA as genomes can be packaged with host histones into the virion. Indeed, this has been described for HSV-1 and some adenoviruses, for example [114,115]. Histone modifications are catalyzed by enzymes that add or delete chemical groups (mainly acetyl and methyl groups) in lysine or arginine residues. For instance, enzymes adding an acetyl group are called histone acetyltransferases (HAT), while the opposite reaction is carried out by histone deacetyltransferases (HDACs) (Table 2) [116]. Importantly, acetyl groups modify the electrostatic charges of histones affecting molecular interactions between the histones and DNA, altering the opening or closure of chromatin structures [116]. Differences in the quantity and types of acetyl groups added to histones are associated with the activation or repression of transcriptional activity [116]. For example, acetylation of lysine 27 in histone 3 (H3K27ac) is associated with active promoters and enhancers because this modification favors open chromatin, and therefore transcriptions factors can bind to promoters to start transcription [117]. While H4K20ac is associated with gene repression in human cells because it promotes chromatin closure [118].

Additionally, histones can also be methylated in a reaction catalyzed by histones methyltransferases (HMT), and different modifications have been described, such as the trimethylation of histone 3, lysine 27 (H3K27me3), which is associated with transcriptional repression, while methylation in H3K4me is a modification related to the transcriptional activity of genes [119].

### DNA Modifications during Viral Infections


**5-methylcytosine (5mC)**


Epigenetic modifications are responsible of phenotypic or gene expression changes that cannot be explained by alterations in DNA sequences [120]. Although these modifications are widely known to occur in the genome of host cells, DNA methylation, particularly 5-methylcytosine (5mC), has also been described in the context of infections with DNA viruses (Table 2) [121]. This modification consists in the addition of a methyl group at the carbon in position 5′ of cytosines, in a reaction that is catalyzed by DNA methyl transferase enzymes (DNMTs) DNMT1 and DNMT3 [120]. This modification mainly occurs at CpG dinucleotides, also called “CpG islands” [120]. Importantly, methylation is a reversible modification and can be edited by demethylase enzymes called ten-eleven translocation methylcytosine dioxygenase (TET), which oxidize 5′methylcytosine to intermediary compounds that are finally converted into cytosine [126]. Noteworthily, 5mC can be recognized by methyl-binding proteins (MBPs), a type of reader protein, which can act as a “scaffold” protein between the modified DNA and repressor complexes, such as Rest Corepressor (CoREST), with gene silencing as a result [127]. 


**Epigenetic Modifications in Alphaherpesvirus Infections**


Although the incoming DNA genome of HSV-1 lacks 5mC, it is methylated by DNMT’s of the host upon infection [122]. Methylation of the HSV-1 genome by DNMT3A, which decreases at later stages of the viral replication cycle, was found to be important for viral replication [122]. Indeed, knocking down of DNMT3A induced a dramatic decrease in viral titers in human fibroblast foreskin-infected cells, highlighting the critical relevance for an active viral replication [122]. Interestingly, during the latency phase of this virus in rat Schwann cells (RSC), CpG methylations were not identified, relating this modification with gene repression, and suggesting that infected cells might use this process to silence viral gene expression [121]. Therefore, methylation of viral DNA occurs during the lytic phase and subsequently seems to be deleted during the establishment of latency [121,128]. Thus, an interesting field for further investigations is how HSV-1 regulates the deletion of methylations during the switch of lytic and latent phases and how viral proteins can regulate the host machinery of DNA methylation during latency.

Another epigenetic modification studied in the context of HSV-1 infections is post-translational modifications in histones. Evidence suggests that the alphaherpesvirus protein US3 from HSV-1, VZV and PrV can hyperphosphorylate histone deacetylase 2 (HDAC2) and that this effect can reduce viral genome silencing during latency phases to allow efficient viral replication upon the lytic phase [129]. Although the HSV-1 genome does not encode histone proteins, 1-h post infection the viral genome has been found to be associated with H3 histones and can actually be packaged into nucleosomes in neurons (SH-SY5Y neuroblastoma cells) [130]. These nucleosomes with H3 are similarly distributed amongst immediate early, early and late HSV-1 genes [130]. Interestingly, an increase in histone modifications that promote DNA availability, such as H3K27ac, were observed to be increased during HSV-1 replication, while repressive modifications, such as H3K27me3, were found to be decreased in THP-1 cells, a monocyte cell line [131]. Therefore, it is possible that the expression of viral genes is regulated by the accessibility of its chromatin [131]. Indeed, the permissive histone modification H3K27ac was found to be critical for viral replication, as its pharmacological inhibition associated with a significant repression of HSV-1 replication [128].

Other studies have found that during early stages of infection with HSV-1, the viral transcriptional co-activator Host cell factor 1 (HCF-1) interacts with different proteins and protein complexes that remodel chromatin, such as methyltransferases and demethylases that specifically modify histones [132,133]. Indeed, during infection with HSV-1 or VZV, HCF-1 was found to interact with remodeling proteins, such as lysine-specific demethylase 1A (LSD1) and mixed-lineage leukemia protein 1 (MLL1), to actively elicit modifications that promote immediate early gene transcription. Furthermore, an interaction between HCF-1 and the chaperone protein anti silencing factor-1 (ASF-1), which manages the flux of histones, was found to be critical for replication of this virus [132].

During latent phases of infection, lytic genes of HSV-1 are condensed into heterochromatin, repressing viral gene expression (Figure 2). This repression has been reported to be associated with the recruitment of a repressor protein complex known as Polycomb repressive complex 1 (PRC1), by proto-oncogene ring finger protein (Bmi1), which negatively impacts the binding of positive regulators of chromatin and promotes modifications such as acetylation of histones by a mechanism unclear to date [134]. Hence, histone modifications associated to viral DNA can modulate the expression of viral transcripts, and during the latent phases these viruses are likely to have a critical role in inhibiting the transcription of viral RNAs [134]. This approach would allow the virus to inhibit host cell antiviral countermeasures, because the genome of HSV-1 is packaged with histones similar to the host DNA [134].

Similar to human herpesvirus, animal alphaherpesviruses, such as MDV, regulate the latent and lytic phases through modifications of the chromatin. Indeed, MDV represses its gene expression in T cells through repressive histone modifications, such as H3K27me3 and H3K9me3 nearby lytic genes in the viral genome [135].

Epigenetic modifications have also been described in other types of alphaherpesviruses, such as VZV. Interestingly, despite the fact that VZV has 61 CpG dinucleotides in its genome, these are not located in promoter regions and therefore, their methylation may not be related to transcriptional suppression. However, more evidence is needed to confirm this hypothesis as 5mC in the body of genes also impact gene expression. Noteworthily, it has been found that during latency after VZV infection, only some viral proteins are expressed, namely the proteins ORF21, ORF29, ORF62 and ORF63 [136]. Importantly, the regulated expression of these ORFs during latency is necessary, and regulation is mediated by epigenetic markers, such as H3K9ac, which are maintained during latency [136,137]. 

Overall, epigenetic modifications have been reported to regulate both the lytic and latent phases of alphaherpesvirus infections (Figure 2). For example, during latency, alphaherpesviruses can condensate their genomes into heterochromatin and only genes related with latency, such as LAT transcripts in HSV-1, and VLT in VZV, are accessible for transcription. Condensed chromatin during latency also allows virus evasion of the immune system, because double-stranded packaged genomes with histones are somewhat similar to host DNA, and because this DNA is episomal and is not producing mRNAs translated into proteins, antiviral responses may not be activated (Figure 2) [138].


**Epigenetic Modifications in Betaherpesvirus Infections**


DNA methylation has been reported in the genome of CMV, specifically within the promoters of immediate early genes being associated with gene repression [123]. Importantly, the pharmacological inhibition of DNA methylation by compounds such as azacytidine have been shown to improve viral replication [123]. This led to the notion that the methylation of viral DNA may be a host defense mechanism for suppressing the transcription of viral genes in infected cells. Interestingly, there is evidence indicating that CMV infection can overall modify the methylation landscape in the infected cell, with a global hypomethylation observed within fibroblasts after infection [123]. Moreover, CMV infection was also found to change the distribution of DNMT1 and DNMT3b from the nucleus to the cytoplasm, with a “DNMT-free environment” in the nucleus being beneficial for virus replication, by evading host mechanisms intended at suppressing viral gene transcription through epigenetic modifications by these enzymes [123].

Other studies reported that CMV replication requires the viral genome to be available for transcription, needing to switch from heterochromatin to euchromatin during productive infection [139]. Accordingly, histone modifications have been described, wherein H3K4me2 has been related to replicating or post-replicative CMV genomes. Other post-translational histone modifications, such as H3K9me2 and H3K9/14ac, have been associated with the viral genome being structured as heterochromatin (Figure 2) [139]. This packaging of the viral genome in heterochromatin and as an episomal form is associated with latency because condensed chromatin is not associated to the production of transcripts nor translation of viral proteins (Figure 2). As mentioned, this would allow the virus to not be recognized by immune system components, avoiding the activation of the host antiviral responses during latency [139].


**Epigenetic Modifications in Gammaherpesvirus Infections**


Similar to the other herpesviruses discussed in this review, the EBV genome has also been found to be methylated in CpG islands, with this modification being associated with transcriptional repression and alternative splicing of viral transcripts [140]. For example, when the cp promoter (first promoter in the EBV genome) is methylated at its upstream sites, a smaller transcript is generated (EBNA 1) and no other EBNA genes are expressed. This gene is related with viral latency in B cells, and the inclusion or exclusion of different EBNA genes is conditioned if latency is established in vivo or in vitro [140,141]. 

Interestingly, an epigenetic mechanism that controls the transition of the EBV virus from a lytic to a latent phase in B cells has been described. During early stages of infection, the transcription factor BZLF1 of this virus is able to bind to 5mC in DNA domains known as meZRE for gene transcription regulation [142]. On the contrary, CpG methylation of these DNA domains was associated with the activation of the transcription of viral lytic genes. Furthermore, genes that did not have CpG methylated dinucleotides were not regulated by BZLF1 [142]. Therefore, DNA methylation of EBV genes may not always be synonymous with gene repression. Importantly, the viral tegument protein (BNRF1) has been reported to prevent histone modifications during early phases of infection and its regulation is indirectly mediated by interactions with the protein Death-associated protein (DAXX), which is associated with chromatin remodelers such as ATRX in an epithelial cell line Hep-2 [124]. The displacement of the interaction between DAXX and ATRX would prevent nucleosome formation with the viral genome, specifically at latent genes [124]. 

Interestingly, inhibition of the expression of BZLF1 has been reported to be needed to maintain EBV latency. In this case, it has been suggested that modifications over histones that limit access to the DNA are critical, particularly with the H3K27me3 and H4K20me3 modifications being involved due to BZLF1 silencing. Also similar to other herpesviruses, EBV uses nucleosome packaging of the viral DNA to maintain latency (Figure 2) [142].

Finally, methylation of the genome of KSHV has been reported, although its methylation pattern did not change up to 6 days post infection in SLK cells. Like other herpesviruses, the genome of KSHV encodes protein factors that can interact with DNMTs enzymes, such as the viral protein LANA, which is involved in the induction and maintenance of latency [143]. This protein also can interact with methyl-binding proteins to suppress genes related to the lytic phase. As KSHV can interact with enzymes involved in epigenetic modifications, it may also modify the methylation pattern of the host cell through the interaction between viral proteins and the cellular methylation machinery, such as DNMT3a and methyl-binding proteins, such as MBP and MeCP. Noteworthily, LANA may use this machinery to suppress the expression of lytic genes [125].

Histone modifications patterns in KSHV infection have also been explored during the latent phase of this virus and its reactivation, wherein repression of lytic genes is reportedly mediated by enzymes of the Polycomb complex, specifically the methylase EZH2, which generates repressive marks in lytic genes to maintain its suppression for maintaining latency in the TRExBCBL1-RTA cell line, which is a cellular line of lymphoma [144]. During reactivation onto the lytic phase, the ORF59 protein has been found to interact with histone demethylases UTX and JMJD3, forming a protein complex with PAN RNA motifs as intermediaries [144]. This interaction allowed the loss of repression modifications in histones, such as H3K27me3, leading to increased expression of viral genes [143,145].

## 5. Concluding Remarks

To date there is no effective treatment for clearing herpesvirus infections or effectively blocking their reactivations in the host. In this review, we revised and discussed how different post-transcriptional RNA modifications are associated with increased or decreased viral fitness, which hints to potential future targets for the development of new drugs that could inhibit RNA-modifying enzymes. However, the development of drugs that can modify RNA or DNA within the host may be double-edged swords, because modifying the epigenome landscape could result in aberrant gene expression patterns in host cells that could result in the malignant transformation of cells.

While the m^6^A is so far the most studied post-transcriptional RNA modification in mRNA and viral RNA, with its presence being described in RNA molecules during alphaherpesvirus, betaherpesvirus and gammaherpesvirus infections, its effect on viral replication would benefit from further studies. Furthermore, a role for this modification will depend on the presence of reader proteins that bind to m^6^A, converting this modification into functional outputs. On the other hand, those reader proteins identified so far in the context of these viral infections, such as YTHDF2, could be modulated positively to have antiviral activity against some herpesviruses, such as beta- and gammaherpesviruses. Moreover, readers such as YTHDF3, which are associated with improved viral replication, could be targeted to dampen viral replication to block the replication cycle of these viruses. Nevertheless, in either case, it is important to consider the effects that such interventions may have over normal cellular functions and thus could be restricted to local applications, particularly over viral infections that act in tissues such as the skin. Still, whether viral proteins have the ability to bind m^6^A-modified RNA is completely unknown. Regarding writer enzymes, such as METTL3, for all viruses discussed in this review these proteins seem to be necessary for viral fitness and this expression is translated in impairments in viral fitness.

The development of high throughput technologies and new antibodies will hopefully allow for advancing this field, to better identify those modifications occurring during infection with herpesviruses and better understand their replication processes. Importantly, while studying RNA modifications is relevant during the infective replication cycle of these viruses, little is known on the role of RNA modifications in the establishment and maintenance of latency, as some viral proteins are not expressed during this process [140]. Naturally, in the context of latency, epigenetic modifications are more likely to play critical roles during this phase, and yet this area has been relatively poorly explored. Interestingly, epigenetic marks such as H3K36me3 were shown to drive m^6^A deposition co-transcriptionally [146]. Whether this crosstalk between epigenetic and epitranscriptomic marks exists during viral infections has never been explored and deserves further exploration.

Although to our knowledge herpesviruses do not encode nucleic-acid-modifying enzymes in their genomes able to write or erase chemical modifications, they encode proteins that can interact with the epigenetic machinery of the host. These interactions allow the regulation of gene expression at different stages of the life cycle of these viruses and, importantly, regulate latency in host cells. Altogether, histone modifications are related with gene regulation and the establishment of latency in host cells.

## Figures and Tables

**Figure 1 microorganisms-10-01754-f001:**
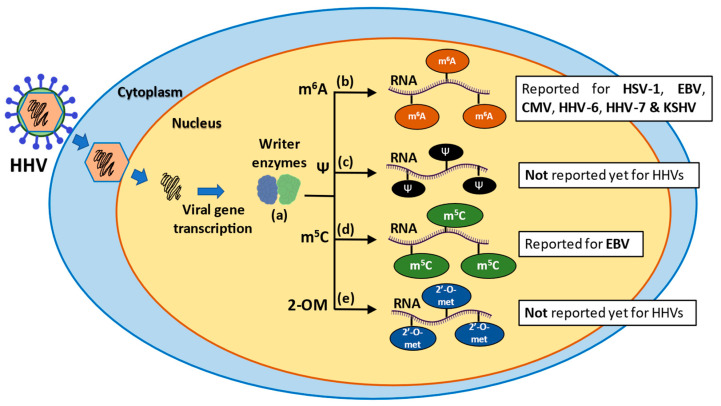
Schematic representation of epigenetic modifications in RNA molecules during human herpesvirus (HHV) infections. (a) After the introduction of the viral DNA into the cytoplasm and posterior transcription of viral genes, writer enzymes perform various epigenetic modifications, such as methyl 6-adenosine (m^6^A), pseudouridylation (Ψ), methyl-5 cytosine (m^5^C) and 2′-O-methylation (2-OM). (b) The m^6^A modification is the most prevalent modification in mRNA and takes place during pre-mRNA synthesis in the nucleus. (c) Pseudouridylation is the most abundant modification in stable RNAs, it is irreversible and occurs through two pathways: an RNA-dependent pathway and RNA-independent pathway. (d) The m^5^C modification is related to improved function and stability of RNA molecules. (e) The 2-OM modification is found in rRNA and tRNA molecules and is associated with increased stability and translation modulation of mRNAs.

**Figure 2 microorganisms-10-01754-f002:**
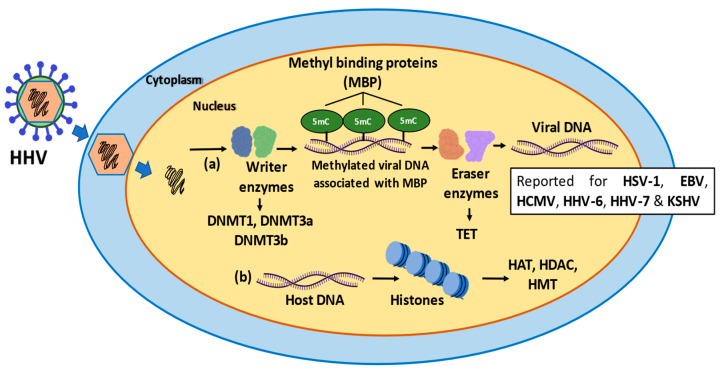
Schematic representation of the epigenetic modifications occurring in viral and host DNA during infection with herpesviruses. (a) Writer enzymes, such as DNMT1, DNMT3a, DNMT3b perform methylations in the viral DNA known as 5’methyl-cytosine (5mC) using SAM as a substrate. The methylation of the DNA is associated with methyl binding proteins (MBP), a type of reader protein, acting as a “scaffold” between 5mC and repressor of complexes such as CoREST. Next, the eraser enzymes (TET) can erase the methylation and form intermediates that are ultimately converted to cytosine such as 5’-hydroxy-methyl-cytosine (5hmC). (b) The histones can also suffer epigenetic modifications by enzymes such as HAT, HDAC and HMT, that favor the latency phase, depending on the modification. Acetylation of lysine 27 in histone 3 (H3K27ac), is associated with active promoters and enhancers. H4K20ac is associated with gene repression. Methylation of lysine 27 in histone 3 is associated with transcriptional repression. H3K4me is related to transcriptional activity of genes.

**Table 1 microorganisms-10-01754-t001:** RNA modifications through writer and eraser enzymes.

	Writers	Erasers	Function	Viruses in Which Its Effects Have Been Described
Methyl6-adenosine (m^6^A)	A complex between METTL3 and METTL14. This protein complex is associated with the WTAP protein.	ALKBH5 and FTO through demethylation reactions.	Regulation of mRNA metabolism and subcellular localization.	HIV, SARS-CoV-2, Zika virus, dengue virus, HCV and HSV-1 [61,62,63,64,65,66,67]
Pseudouridylation (Ψ)	RNA-dependent: H/ACA sRNPs using a complementary gRNA. RNA-independent: PUS.	Unidentified.	Regulation of alternative splicing and the translation efficiency of mRNAs. Associated also with immune escape.	SARS-CoV-2 [68,71]
Methyl 5-cytosine (m^5^C)	NSUN 1,2,3,4,5,6,7.	Unidentified.	Function and stability of mRNAs.	SARS-CoV-2, HIV, EBV [80,81,82,83,84]
2’-O-Methylation (2-OM)	MAT and some viral proteins (NSP10/14 and NSP10/16).	Unidentified.	It is associated with RNA stability and translation modulation.	HIV, West Nile virus, SARS-CoV-2 [88,90].

**Table 2 microorganisms-10-01754-t002:** Viral DNA modifications by writer and eraser enzymes.

	Writers	Erasers	Function	Viruses in Which Effects Have Been Described
**5 methyl-cytosine (5mC)**	DNMTs (1, 3a, 3b)	TET enzymes.	Associated with gene repression.	HSV-1, EBV, HCMV, HHV-6, HHV-7, KSHV [120,121,122].
**Histone modifications**	HAT, HMT, trymethylation	HDAC, HDMT.	Diverse modifications are involved in gene regulation.	HSV-1, HSV-2, VZV, EBV, HCMV, HHV-6, HHV-7, KSHV [121,123,124,125].

## Data Availability

Data sharing does not apply to this article as no new data were created or analyzed in this review.

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
