# Peer review of "Role of Epitranscriptomic and Epigenetic Modifications during the Lytic and Latent Phases of Herpesvirus Infections"

_microorganisms, 2022, doi:10.3390/microorganisms10091754_

Round 1
Reviewer 2 Report
In this review, Soto and co-workers reported a study on the role of viral RNA and DNA modifications during the lytic and latent phases of herpesvirus infections, reviewed and discussed current evidence regarding epitranscriptomic and epigenetic modifications of herpesviruses, and how these modifications affect their life cycle. The authors discussed how different post-transcriptional RNA modifications are associated with increases or decreases in viral fitness, hinting at potential future targets for the development of new drugs. Overall, the idea of the paper is clear, but it needs to be improved before it can be accepted for publication.
1. The order of the authors' writing is Introduction, Herpesviruses: Epidemiology and illnesses, epitranscriptomic modifications and herpesviruses and DNA epigenetic modifications during in viral infections, where all the headings are without numbers except "1. Introduction ". For a more logical and coherent article, I suggest that the headings be adjusted as follows: 1. Introduction, 2. Herpesviruses: Epidemiology and illnesses 3. epitranscriptomic modifications and herpesviruses, 3.1 Episodic splicing modifications, 3.2 Episodic transcriptional modifications in herpesvirus infection 4. DNA epigenetic modifications during in viral infections, 4.1 Epigenetic modifications of DNA 4.2 Epigenetic modifications in herpesvirus infection.
2. lines 66-72, the authors claimed temporal specificity of gene expression during herpesvirus HSV-1 infestation of host cells, starting with immediate early genes, early genes, and late (early-late, post-late) genes. I think it is appropriate to add the characteristics and criteria for delineation of its various temporal genes here, and give examples. Just as ICP4 gives examples of specific genes expressed by HSV-1 in each period.
3. For the description of epigenetic splicing modification and DNA epigenetic modification viruses, it is suggested to add some herpes viruses such as Pseudorabies virus (PRV), and Marek's disease virus (MDV), etc.
4. The authors explained the two RNA-dependent and RNA-independent pathways of pseudouridine at 190-200. RNA-independent RNA can modify tRNA (PUS 2/3/4/6/9), small nucleolar RNA (PUS 1/7), rRNA (PUS 5/7), and mRNA (PUS 1/2/3/4/6/7/9), but the types that can be modified by RNA-dependent pathway modifications are not provided.
5. Lines 72-85, regarding the contribution of HSV glycoproteins to infected cells, I think this description should be moved to lines 60-61, “For HSV-1 and HSV-2, then gD mediates the interaction between the virus and additional cell receptors after the virus has adhered to the host cell surface”. In addition, the epidemiological descriptions of VZV, CMV, HHV-6, HHV-7, EBV, and KSHV mainly focused on virus name, infection rate, mode of transmission, symptoms of infection, distribution status, and cell type of infection, but no detailed description of the cellular mechanism of infection appeared, which is recommended to be added.
6. In lines 251-256, the authors described the contribution of 2'-O-methylation in virus evasion of intrinsic cellular defense mechanisms, but the examples cited do not support this view.
7. Firstly, the authors should reformat Table.1 and Table. 2 to make them easier to view, especially regarding the function. Secondly, the description in the function column reads: “Adds a methyl group at position N6 of adenosine to form m6A. It is the most prevalent internal modification in RNAs. It is the most prevalent internal modification in RNAs”, which is not a functional description of methylation, but more like a definition, please summarize the function for modification.
8. In the keywords, "Latency" should be followed by a semicolon instead of a comma.
9. The author describes several times in the text that herpes viruses disperse RNA methylesterase from the nucleus to the cytoplasm, if possible, please provide in detail the mechanism by which this process occurs.
10. The abbreviations appear multiple times in the text without giving the full name, such as SAM, HMT, and HCV.
11. Lines 314-315, “Interestingly, CMV infection of NK cells affects in different manners the proteins that read or erase the m6A modification in the host cell.” No detailed mechanism of action is given, and the examples given only demonstrate the results that can be influenced, please add details about the mechanisms of influence in different ways.
Author Response
In this review, Soto and co-workers reported a study on the role of viral RNA and DNA modifications during the lytic and latent phases of herpesvirus infections, reviewed and discussed current evidence regarding epitranscriptomic and epigenetic modifications of herpesviruses, and how these modifications affect their life cycle. The authors discussed how different post-transcriptional RNA modifications are associated with increases or decreases in viral fitness, hinting at potential future targets for the development of new drugs. Overall, the idea of the paper is clear, but it needs to be improved before it can be accepted for publication.
The order of the authors' writing is Introduction, Herpesviruses: Epidemiology and illnesses, epitranscriptomic modifications and herpesviruses and DNA epigenetic modifications during in viral infections, where all the headings are without numbers except "1. Introduction ". For a more logical and coherent article, I suggest that the headings be adjusted as follows: 1. Introduction, 2. Herpesviruses: Epidemiology and illnesses 3. epitranscriptomic modifications and herpesviruses, 3.1 Episodic splicing modifications, 3.2 Episodic transcriptional modifications in herpesvirus infection 4. DNA epigenetic modifications during in viral infections, 4.1 Epigenetic modifications of DNA 4.2 Epigenetic modifications in herpesvirus infection.
Response: Thank you for this comment. We have adopted the structure suggested by the Reviewer with the corresponding headings.
Reviewer: lines 66-72, the authors claimed temporal specificity of gene expression during herpesvirus HSV-1 infestation of host cells, starting with immediate early genes, early genes, and late (early-late, post-late) genes. I think it is appropriate to add the characteristics and criteria for delineation of its various temporal genes here, and give examples. Just as ICP4 gives examples of specific genes expressed by HSV-1 in each period.
Response: Thank you for this comment. We have added in the corresponding section some examples of viral genes of each type (Immediate Early, Early and Late) and a brief explanation regarding the criteria for classification.
Reviewer: For the description of epigenetic splicing modification and DNA epigenetic modification viruses, it is suggested to add some herpes viruses such as Pseudorabies virus (PRV), and Marek's disease virus (MDV), etc.
Response: As suggested by the Reviewer, we have added the corresponding new sections and viruses to the revised version of the manuscript.
Reviewer: The authors explained the two RNA-dependent and RNA-independent pathways of pseudouridine at 190-200. RNA-independent RNA can modify tRNA (PUS 2/3/4/6/9), small nucleolar RNA (PUS 1/7), rRNA (PUS 5/7), and mRNA (PUS 1/2/3/4/6/7/9), but the types that can be modified by RNA-dependent pathway modifications are not provided.
Response: Thank you for this observation. We have edited the text to better describe RNA-dependent pathway modifications at the corresponding section.
Reviewer: Lines 72-85, regarding the contribution of HSV glycoproteins to infected cells, I think this description should be moved to lines 60-61, “For HSV-1 and HSV-2, then gD mediates the interaction between the virus and additional cell receptors after the virus has adhered to the host cell surface”. In addition, the epidemiological descriptions of VZV, CMV, HHV-6, HHV-7, EBV, and KSHV mainly focused on virus name, infection rate, mode of transmission, symptoms of infection, distribution status, and cell type of infection, but no detailed description of the cellular mechanism of infection appeared, which is recommended to be added.
Response: Thank you for this comment. We have added a brief description regarding infection at the cellular level for each virus mentioned in the text.
Reviewer: In lines 251-256, the authors described the contribution of 2'-O-methylation in virus evasion of intrinsic cellular defense mechanisms, but the examples cited do not support this view.
Response: Thank you for this observation. We have corrected the reference intended to be cited at this location.
Reviewer: Firstly, the authors should reformat Table.1 and Table. 2 to make them easier to view, especially regarding the function. Secondly, the description in the function column reads: “Adds a methyl group at position N6 of adenosine to form m6A. It is the most prevalent internal modification in RNAs. It is the most prevalent internal modification in RNAs”, which is not a functional description of methylation, but more like a definition, please summarize the function for modification.
Response: Thank you for the suggestions made for the Tables, For ease of view, we have merged the reference column with the column indicating which viruses these observations have been made. Also, we have edited the “Function” column to better describe the functions of each modification.
Reviewer: In the keywords, "Latency" should be followed by a semicolon instead of a comma.
Response: Thank you for this observation. We have corrected the indicated typo in the revised updated manuscript.
Reviewer: The author describes several times in the text that herpes viruses disperse RNA methylesterase from the nucleus to the cytoplasm, if possible, please provide in detail the mechanism by which this process occurs.
Response: Thank you for this observation. We have edited the text to better describe the mechanism by which viral proteins relocate RNA methyltransferases at the corresponding section.
Reviewer: The abbreviations appear multiple times in the text without giving the full name, such as SAM, HMT, and HCV.
Response: Thank you for this observation . We added the full name to the indiacted abbreviations in the revised manuscript.
Reviewer: Lines 314-315, “Interestingly, CMV infection of NK cells affects in different manners the proteins that read or erase the m6A modification in the host cell.” No detailed mechanism of action is given, and the examples given only demonstrate the results that can be influenced, please add details about the mechanisms of influence in different ways.
Response: Thank you for observation. We have edited the test to indicate what is known to date regarding the reported effect.
Once again, we would like to thank the Reviewer for the valuable time and effort invested in revising our manuscript. We feel that the updated version of the article has significantly improved after this revision.

Reviewer 3 Report
1. Generally, preparing such a review on epitranscriptomic and epigenetic modifications of nucleic acids during herpesvirus infection is justified as there are no recent reviews covering all the herpesvirus subfamilies in one paper. The paper is well written and my remarks are rather editorial.
T The structure of the review contains a description of herpesviruses, herpesvirus-related diseases, and their prevalence in human population, then it describes the main RNA (and DNA) modifications as separate chapters, modifications in general, shortly introducing modifications in viruses and describing the data on modifications in herpesvirus infections in separate chapters. The review focuses clearly on human pathogens, but I miss here one sentence mentioning that numerous herpesviruses infect animals other than human (especially since PRV is mentioned later on in the text). Collecting recent data on the prevalence of herpesviruses in the human population (in %) is beneficial, it can be used as a source of quotations for other papers.
I recommend re-thinking the title of this review. For sure, it does not cover all the important RNA and DNA modifications during herpesvirus infections. E.g., the role of APOBEC and DNA cytosine deamination is not covered. APOBECs are, on the other hand, described in another fresh review, in MDPI Viruses. I think it would be better to make the title more detailed, like including the words epigenetic, epitranscriptomic, as in the abstract. If epitranscriptomic and epigenetic appear in the title, the list of keywords could be also changed as for now it is very general and does not allow to track the paper well. Maybe the keywords, like m6A, 5mC, histone modifications should be included.
Please work on the list of references. Include all the authors (naming only the first author et al. is not acceptable, some titles have small and capital letters mixed (like the title starts with a small letter but contains capital letters inside).
Please insert spaces before reference numbers in brackets[x].
The column Function in both Tables, especially Table 2 is very narrow and hard to read, please reconsider editing the Tables for better visibility of the text.
Line 73: “numerous glycoproteins on the outer surface”, please note that herpesvirus envelopes contain not only glycoproteins but also other proteins, with no sugar modifications.
Line 132: EBV is known to be an oncogenic virus, as its infection is associated with Hodgkin’s lymphoma: EBV is associated with several cancers, not only Hodgin’s lymphoma, so add “mainly” or include other EBV-related cancers.
Line 168: unclear, consists of?
Line 173: SAM is mentioned for the first time. Please explain the abbreviation.
Line 193: The name of the RNPs in brackets looks strange.
Line 195: pseudouridine
Line 220: TLR3 is more common than TLR-3.Similarly, line 371: TLR9
Line 255: I think the full name of IFIT proteins includes tetratricopeptide
Line 311: in betaherpesviruses infections – should be betaherpesvirus infections
Line 327: Add a coma after METTL3
Legend of Figure 1: Adding and explaining HHV abbreviation is recommended.
Line 391: 11 after transcripts?
Line 392: I would expand to colleagues, looks nicer.
Line 394: Please add what type of cells are iSLK
Like 477: Although the HSV-1 genome does not contain histone proteins – this is unclear: does not encode?
Line 478: The full name of this cell line is SH-SY5Y.
Line 596: Why BZLF1 in brackets?
Line 600: Please mention what type of cells are Hep-2.
Line 630: there are no – there is no
Author Response
1. Generally, preparing such a review on epitranscriptomic and epigenetic modifications of nucleic acids during herpesvirus infection is justified as there are no recent reviews covering all the herpesvirus subfamilies in one paper. The paper is well written and my remarks are rather editorial.
The structure of the review contains a description of herpesviruses, herpesvirus-related diseases, and their prevalence in human population, then it describes the main RNA (and DNA) modifications as separate chapters, modifications in general, shortly introducing modifications in viruses and describing the data on modifications in herpesvirus infections in separate chapters. The review focuses clearly on human pathogens, but I miss here one sentence mentioning that numerous herpesviruses infect animals other than human (especially since PRV is mentioned later on in the text). Collecting recent data on the prevalence of herpesviruses in the human population (in %) is beneficial, it can be used as a source of quotations for other papers.
I recommend re-thinking the title of this review. For sure, it does not cover all the important RNA and DNA modifications during herpesvirus infections. E.g., the role of APOBEC and DNA cytosine deamination is not covered. APOBECs are, on the other hand, described in another fresh review, in MDPI Viruses. I think it would be better to make the title more detailed, like including the words epigenetic, epitranscriptomic, as in the abstract. If epitranscriptomic and epigenetic appear in the title, the list of keywords could be also changed as for now it is very general and does not allow to track the paper well. Maybe the keywords, like m6A, 5mC, histone modifications should be included.
Response: Thank you for these comments and recommendations. As suggested, we have edited the title of the manuscript to better describe the areas being revised. Additionally, we have edited the introduction to add data on the prevalence of the herpesviruses revised.
Reviewer: Please work on the list of references. Include all the authors (naming only the first author et al. is not acceptable, some titles have small and capital letters mixed (like the title starts with a small letter but contains capital letters inside).
Response: Thank you for this observation. We have revised and edited the references that needed corrections.
Reviewer: Please insert spaces before reference numbers in brackets[x].
Response: Thank you for this comment. We have inserted a space between the last character in a sentence and the corresponding reference number.
Reviewer: The column Function in both Tables, especially Table 2 is very narrow and hard to read, please reconsider editing the Tables for better visibility of the text.
Response: As requested, we have edited the table for better visibility. The columns with the references was merged with the column indicating in which viruses these outlined aspects have been observed.
Reviewer: Line 73: “numerous glycoproteins on the outer surface”, please note that herpesvirus envelopes contain not only glycoproteins but also other proteins, with no sugar modifications.
Response: Thank you for this observation. We have amended the indicated text with “proteins and glycoproteins”.
Reviewer: Line 132: EBV is known to be an oncogenic virus, as its infection is associated with Hodgkin’s lymphoma: EBV is associated with several cancers, not only Hodgin’s lymphoma, so add “mainly” or include other EBV-related cancers.
Response: Thank you for this comment. We have edited the text adding “mainly” as suggested.
Reviewer: Line 168: unclear, consists of?
Response: Thank you for this observation. We have edited the text at this location for a better understanding of the content.
Response: We have modified the text for a better understanding.
Reviewer: Line 173: SAM is mentioned for the first time. Please explain the abbreviation.
Response: Thank you for this observation. We have added the meaning of this abbreviation at this location in the text.
Reviewer: Line 193: The name of the RNPs in brackets looks strange.
Response: Thank you for your suggestion. We have removed the brackets in this sentence.
Reviewer: Line 195: pseudouridine
Response: Thank for this observation, we corrected the orthographic error.
Reviewer: Line 220: TLR3 is more common than TLR-3. Similarly, line 371: TLR9
Response: Thank you for suggestions. We modify “TLR-3” and “TLR-9” to “TLR3” and “TLR9” in the revised manuscript.
Reviewer: Line 255: I think the full name of IFIT proteins includes tetratricopeptide
Response: Thank you for this observation. We have corrected the definition of the abbreviation of IFIT in the text.
Reviewer: Line 311: in betaherpesviruses infections – should be betaherpesvirus infections
Response: Thank you for this observation. We have amended the text accordingly.
Reviewer: Line 327: Add a coma after METTL3
Response: Thank you for this observation, we have added a coma after METTL3 at the indicated line.
Reviewer: Legend of Figure 1: Adding and explaining HHV abbreviation is recommended.
Response: Thank you for this suggestion. We have added a description for the HHV abbreviation in the legend of Figure 1.
Reviewer: Line 391: 11 after transcripts?
Response: Thank you for this observation. We have corrected the typo at the indicated line.
Reviewer: Line 392: I would expand to colleagues, looks nicer.
Response: Thank you for suggestion. Modification performed.
Reviewer: Line 394: Please add what type of cells are iSLK
Response: Thank you for this observation. We have added a description regarding what type of cells iSLK are.
Reviewer: Like 477: Although the HSV-1 genome does not contain histone proteins – this is unclear: does not encode?
Response: Thank you for this observation. We have edited the text to clarify that the HSV-1 genome does not “encode” histone proteins.
Reviewer: Line 478: The full name of this cell line is SH-SY5Y.
Response: Thank you for this observation. We have corrected the orthographic error.
Reviewer: Line 596: Why BZLF1 in brackets?
Response: Thank you for this observation. We have removed the brackets at the indicated line.
Reviewer: Line 600: Please mention what type of cells are Hep-2.
Response: Thank you for this observation. We added to the text a brief description indicating what type of cells are Hep-2.
Reviewer: Line 630: there are no – there is no
Response: Thank you for observation. We corrected the text accordingly.
Once again, we would like to thank the Reviewer for the valuable time and effort invested in revising our manuscript. We feel that the updated version of the article has significantly improved after this revision.
